# Population Pharmacokinetics of Vincristine Related to Infusion Duration and Peripheral Neuropathy in Pediatric Oncology Patients

**DOI:** 10.3390/cancers12071789

**Published:** 2020-07-04

**Authors:** Mirjam Esther van de Velde, John Carl. Panetta, Abraham J. Wilhelm, Marleen H. van den Berg, Inge M. van der Sluis, Cor van den Bos, Floor C.H. Abbink, Marry M. van den Heuvel-Eibrink, Heidi Segers, Christophe Chantrain, Jutte van der Werff Ten Bosch, Leen Willems, William E. Evans, Gertjan L. Kaspers

**Affiliations:** 1Emma Children’s Hospital, Amsterdam University Medical Center, Vrije Universiteit Amsterdam, Pediatric Oncology, 1081 HV Amsterdam, The Netherlands; mh.vandenberg@amsterdamumc.nl (M.H.v.d.B.); gjl.kaspers@amsterdamumc.nl (G.L.K.); 2Department of Pharmaceutical Sciences, St. Jude Children’s Research Hospital, Memphis, TN 38105, USA; carl.panetta@stjude.org (J.C.P.); william.evans@stjude.org (W.E.E.); 3Amsterdam University Medical Center, Vrije Universiteit Amsterdam, Department of Clinical Pharmacology, 1081 HV Amsterdam, The Netherlands; aj.wilhelm@amsterdamumc.nl; 4Princess Máxima Center for Pediatric Oncology, 3584 CS Utrecht, The Netherlands; I.M.vanderSluis@prinsesmaximacentrum.nl (I.M.v.d.S.); C.vandenBos-5@prinsesmaximacentrum.nl (C.v.d.B.); m.m.vandenheuvel-eibrink@prinsesmaximacentrum.nl (M.M.v.d.H.-E.); 5Emma Children’s Hospital, Amsterdam University Medical Center, Amsterdam Medical Center, Pediatric Oncology, 1105 AZ Amsterdam, The Netherlands; f.abbink@amsterdamumc.nl; 6Department of Pediatric Hemato-Oncology, University Hospitals Leuven, 3000 Leuven, Belgium; heidi.segers@uzleuven.be; 7Department of Pediatrics, Clinique du MontLégia, CHC, 4000 Liège, Belgium; christophe.chantrain@chc.be; 8Department of Pediatric Onco-Hematology, Universitair Ziekenhuis Brussel, 1090 Brussels, Belgium; jutte.VanderWerffTenBosch@uzbrussel.be; 9Department of Paediatric Haematology-Oncology and Stem Cell Transplantation, Ghent University Hospital, 9000 Ghent, Belgium; leen.willems@uzgent.be

**Keywords:** neurotoxicity, exposure, children, cancer, vincristine, toxicity, administration duration, infusion rate, adolescent, chemotherapeutic, oncovin

## Abstract

Vincristine (VCR) is frequently used in pediatric oncology and can be administered intravenously through push injections or 1 h infusions. The effects of administration duration on population pharmacokinetics (PK) are unknown. We described PK differences related to administration duration and the relation between PK and VCR-induced peripheral neuropathy (VIPN). PK was assessed in 1–5 occasions (1–8 samples in 24 h per occasion). Samples were analyzed using high-performance liquid chromatography/tandem mass spectrometry. Population PK of VCR and its relationship with administration duration was determined using a non-linear mixed effect. We estimated individual post-hoc parameters: area under the concentration time curve (AUC) and maximum concentration (C_max_) in the plasma and peripheral compartment. VIPN was assessed using Common Terminology Criteria for Adverse Events (CTCAE) and the pediatric-modified total neuropathy score (ped-mTNS). Overall, 70 PK assessments in 35 children were evaluated. The population estimated that the intercompartmental clearance (IC-Cl), volume of the peripheral compartment (V_2_), and C_max_ were significantly higher in the push group. Furthermore, higher IC-Cl was significantly correlated with VIPN development. Administration of VCR by push led to increased IC-Cl, V_2_, and C_max_, but were similar to AUC, compared to 1 h infusions. Administration of VCR by 1 h infusions led to similar or higher exposure of VCR without increasing VIPN.

## 1. Introduction

Vincristine (VCR) is a frequently used chemotherapeutic agent for the treatment of several types of pediatric malignancies [1]. It works by inhibiting mitosis [2,3], and VCR is primarily eliminated by metabolism in the liver via the cytochrome P450 (CYP) enzymes, particularly CYP3A4 and CYP3A5 [4].

The development of a high-performance liquid chromatographic (HPLC) assay for VCR [5] has generated reliable data regarding the pharmacokinetics (PK) of VCR in children. Previous studies have shown a large inter- and intra-patient PK variability [6,7,8,9,10] and a higher clearance (Cl) rate in children compared to adults [6].

Previous studies investigating the relationship between VCR PK and toxicity showed conflicting results, with some studies indicating that there is no association between PK and VCR-induced peripheral neuropathy (VIPN) [6,8,11,12], the major dose-limiting toxicity of VCR, whereas others did report this association [1,13,14].

VIPN is caused by the disruption of microtubules within the cell, resulting in neuronal axon dysfunction. Inflammatory processes also seem to play a role in the pathophysiology of VIPN. Ultimately, it leads to a progressive motor and sensory and autonomic nerve damage due to the dysfunction of Aβ, Aδ, and C-fibers [14]. VIPN can lead to dose reductions or omitted doses [1,15,16] of VCR. In children, VCR doses higher than 2 mg/m^2^ (with a maximum of 2 mg) have shown to lead to intolerable VIPN [15]. However, Frost et al. suggested that the common practice of dose-capping VCR at 2 or 2.5 mg should be carefully reconsidered, since they found no evidence to suggest age-dependent PK of VCR, indicating that by capping the VCR dose, older children are less intensely treated than younger children [17]. The duration of VCR administration may also play a role in the development of VIPN. When continuous administrations of up to 5 consecutive days were used, doses up to 7.5 mg/m^2^ without dose-capping were well tolerated in children and adults [18,19]. However, it is evident that such long administration durations and high doses are costly and cumbersome and there is no clear indication of a better therapeutic efficacy [20]. Therefore, an alternative 1 h infusion is more feasible and may also offer a favorable toxicity profile compared to shorter administration durations. However, in clinical practice, very short administration times, i.e., push injections lasting 1–5 min, are often the standard care for pediatric cancer patients. Administration time directly influences the maximum concentration (C_max_), which was previously associated with VIPN [21]. However, the effect of administration duration of VCR on its PK remains to be established, as well as how these PK are related to the development of VIPN in children.

In addition, a drug–drug interaction between VCR and concurrent treatment with many azole antifungals was previously reported [22,23,24]. These azole antifungals are frequently used during intensive chemotherapeutic treatment in children for the prevention or treatment of invasive fungal infections [25]. The interaction is based on the inhibiting effect of azole antifungals on CYP3A4 and P-glycoprotein [26]. However, possible associations between this drug–drug interaction and VCR PK are largely unknown [27].

To address these issues, the primary aims of the current study were: (1) to assess PK of VCR given to pediatric cancer patients as either push injections or 1 h infusions; (2) to investigate the association between PK and exposure parameters and VIPN; (3) to investigate the effect of concurrent azole antifungal treatment on VCR PK parameters.

## 2. Results

### 2.1. Study Population

In total, 35 out of 90 patients enrolled in a randomized clinical trial (RCT) and also participated in the PK studies. Table 1 shows the characteristics of these 35 patients, which represented the characteristics of the entire cohort of 90 patients. In total, 20 patients (57%) received their VCR through push injections, while 1 h infusions were given in 15 patients (43%). All patients in the study received the standardized dose of 1.5 mg/m^2^ or 2 mg/m^2^ with a maximum of 2 mg. In total, PK measurements were performed on 70 different occasions (1–5 per patient), which resulted in a total of 425 samples. Doses were capped at 2 mg in 20 patients (37 occasions). At two of the 70 occasions (3%), two patients received a 15-min VCR infusion instead of the administration method, according to their randomization. These were analyzed as push injections.

### 2.2. Population Pharmacokinetics of VCR

The population PK for VCR are summarized in Table 2 and Figure 1. First, we considered the effects of the VCR administration method (push vs 1 h infusion). We observed that there were significant differences in the PK parameters that affect the terminal phase. Specifically, intercompartmental clearance (IC-Cl) and volume of the peripheral compartment (V_2_) were each 3.1 times higher in the push vs 1 h infusion group (*p*-5.1 × 10^−9^; Table 2). While there was a trend to lower VCR Cl in the push group, this difference did not reach significance (*p* = 0.058). The individual weighted residuals of our final model can be found in Figure 2.

We also considered the effects of the azole antifungal treatment on the VCR PK but did not find any significant relationships.

In addition, we evaluated differences in VCR exposure due to the administration method. We observed a higher, although not statistically significant, area under the concentration time curve (AUC) in the 1 h infusion group (median plasma AUC 1 h group: 44.04 (ng·hr)/mL, median plasma AUC push group: 38.60 (ng·h)/mL, *p* = 0.22, median peripheral AUC 1 h group: 42.50 (ng·h)/mL, median peripheral AUC push group: 35.36 (ng·h)/mL, Table 3). In addition, we observed significantly higher plasma C_max_ but significantly lower peripheral C_max_ in the push group compared to the 1 h group (median plasma C_max_ > 2 times higher in the push group; *p* = <0.001, median peripheral C_max_ 2 times lower in the push group; *p* < 0.001, Table 3). Finally, the median time, with a plasma VCR concentration above 1 ng/mL, differed between the two administration groups: 1- h group median time = 0.92 h, push group median time = 0.24 h; *p* < 0.001, as well as the median time with an estimated peripheral VCR concentration above 1 ng/mL (1 h median time: = 14.65 h, push median time = 16.83 h; *p* = 0.001).

### 2.3. Association between PK Parameters and VIPN

Both scores for the total Common Terminology Criteria for Adverse Events (CTCAE) and pediatric-modified total neuropathy score (ped-mTNS) and the scores for the dichotomized CTCAE and ped-mTNS were significantly associated with IC-Cl, with higher IC-Cl associated with VIPN (see Table 4). All other primary parameters (Cl, volume of the central compartment (V_1_) and (V_2)_), as well as post-hoc parameters (plasma AUC and C_max_ and peripheral AUC and C_max_) were not associated with any of the VIPN outcomes.

## 3. Discussion

This study evaluated the effects of administration duration (push injections versus 1 h infusions) on the pharmacokinetics of VCR in children with cancer, along with the effects of VCR PK on VIPN. We observed that the intercompartmental clearance (IC-Cl), volume of the peripheral compartment (V_2_), and plasma and peripheral C_max_ were significantly higher in the push group compared to individuals in the 1 h group. Finally, we showed that higher IC-Cl was significantly correlated with VIPN development. 

When VCR PK was previously studied, VCR was generally administered as a push injection. In these studies, the mean Cl of VCR push injections varied between 13.44 and 28.9 L/h/m^2^ [6,7,8,17,20], whereas Cl of our push cohort had a mean (range) of: 26.86 (13.45–48.40) L/h/m^2^. However, the mean Cl of the 1 h cohort was (range): 32.61 (19.76–72.67) L/h/m^2^, which is higher than previously published data of Cl in push injected VCR and corresponds well with our nearly statistically significant finding of higher Cl in the 1 h group compared to the push injection group. In addition, in our push cohort median plasma AUC was slightly lower than previously published median plasma AUC (46.17–111.89 (ng·h)/mL [7,17,20]); however, our range of measured AUC’s [17.22–63.05 (ng·h)/mL] was in-line with these other median values of AUC. One previous VCR PK study by Guilhaumou et al. [8] (VCR administered as push injection) reported values of IC-Cl (51.9 L/h/m^2^), V_1_ (15.9 L/m^2^), and V_2_ (145 L/m^2^), which differed from the results in our push cohort: median (range): IC-CL: 129.73 (30.76–301.56) L/h/m^2^; V_1_: 19.57 (9.03–34.97) L/m^2^; and V_2_: 445.57 (115.08–803.58) L/m^2^.

To the best of our knowledge, this is the first study that aimed to establish the association between administration duration of VCR and PK parameters. However, there are multiple studies that have investigated the effect of prolongation of administration duration on PK in other drugs. One study reported the effects of 1-min versus 1 h and 3 h administrations of midazolam. Just like VCR, midazolam is metabolized in the liver through the CYP3A4 family of enzymes [28]. Results of this study showed that Cl was significantly higher when administered in 1-min compared to an administration time of 1 h or 3 h (*p* < 0.001) [29]. In our study, we also found a nearly statistically significant difference in Cl related to administration duration and a significant difference of IC-Cl. The effect of administration duration on PK measures was also studied in children using ifosfamide, another CYP3A-mediated agent [30]. In this study, continuous infusions (72 h) were compared with 1 h infusions. A higher Cl of ifosfamide was shown in the continuous infusion group compared to the 1 h group, whereas we observed no such difference. However, AUC was similar between the two administration groups [31], which was comparable to our results.

In our study, we observed that higher IC-Cl was significantly associated with the development of VIPN. In previous studies, as mentioned earlier, results regarding the association between PK parameters of VCR and VIPN were conflicting [6,11,12,13,14,32]. In one study, a positive association between VCR AUC and the concentration of metabolite M1 and VIPN was identified. Unfortunately, it is unknown what the VCR administration duration was in this trial [13]. However, in the study by Crom et al., in which VCR was administered as a push injection, no significant association between VCR AUC and VIPN was found [6]. In addition, Moore et al. also found no significant associations between PK parameters and VIPN. In this study, VCR was administered for 5–10 min [11]. Guilhaumou et al. studied IC-Cl, terminal half-life, Cl, and AUC of VCR push injections, in relation to VIPN, and also found none of these parameters to be significantly associated [32]. Finally, Plasschaert et al. studied the relation between Cl, AUC, and distribution and elimination half-life of VCR push injections and constipation, as a marker for VIPN, and did not find any significant associations [12]. All these studies that did not find a significant association between PK and VIPN are in contrast to our findings. This could be due to the fact that we used two different instruments to prospectively assess VIPN. In particular, ped-mTNS was not available when the other studies were performed. ped-mTNS is considered a more sensitive tool in assessing VIPN in children than the commonly used CTCAE criteria [33]. Therefore, we may have detected more subtle symptoms of VIPN compared to these previous studies and thus were able to find an association between IC-Cl and VIPN. Furthermore, in four out of six of these studies, the assessment of VIPN was done retrospectively through medical chart examination [6,11,12,13], whereas we assessed VIPN prospectively.

It has been previously reported that azole antifungal treatment concurrent with VCR dosing influences the development of VIPN [22,23,24]. However, our study does not show any relationship between VCR PK and azole treatment. This is possibly due to our limited sample size (only 7/70 (*n* = 6 patients) measurements were with concurrent azole antifungal treatment). Furthermore, we used any type of azole antifungal to study the relationship with concurrent VCR use, even though there are differences in the CYP450 inhibiting effect between different azole antifungals and, therefore, effects of concurrent VCR-azole antifungal use should ideally be studied for each different type of azole antifungal. Previously, Hasselt et al. [34] reported that a minimum sample size of 38 patients with at least 150 samples was required to detect a Cl inhibition of concurrent azole and VCR treatment with 80% power. This suggests that our data are not powered to detect significant changes in VCR PK due to concurrent azole treatment.

We acknowledge some limitations to this study. First of all, our patient participation was limited compared to our entire cohort, which was probably due to the fact that sampling could not be done by capillary blood drawings or in many occasions through a central line, since this line was used for the administration of VCR in the 1 h group during sampling. Therefore, an extra peripheral cannula had to be inserted for trial participation. This was invasive and led to refusal of the PK part of our study for some participants and withdrawing of informed consent during the study, thereby lowering our participating patient numbers. Despite these dropouts, our sample size remained reasonably large. Secondly, during a few PK sampling occasions, different VCR administration durations were applied in one of the participating hospitals, but we allowed different administration durations within the final covariate model.

## 4. Materials and Methods

### 4.1. Patients

The current study is part of an RCT investigating the relation between administration method of VCR and the development of VIPN in children with cancer. Participants of this RCT received all planned VCR administrations of their treatment protocol either through intravenous push injections or through 1 h infusions. Children with the following diseases and treatment protocols were eligible for participation: acute lymphoblastic leukemia (ALL) (DCOG ALL-11 protocol [34], EORTC-58081-CLG guideline [35], or EsPhALL protocol [36]), Hodgkin lymphoma (EuroNet-PHL-C1 protocol [37] or C2 protocol [38]), rhabdomyosarcoma (EpSSG RMS 2005 protocol [39]), nephroblastoma (SIOP Wilms 2001 protocol [40]), medulloblastoma (ACNS0331 [41] or ACNS0332 [42] protocol), and low-grade glioma (SIOP LGG 2004 protocol [43]). Results of the RCT part of the trial will be reported separately. This trial was registered in the Dutch Trial Registry (www.trialregister.nl) with number NL4019.

Participants of the RCT could choose whether they also wanted to participate in the PK part of the study. Of those who agreed to participate, written informed consent was obtained from parents and/or children (in case the child was ≥12 years). Blood samples were collected between September 2014 and April 2018 in either one of the four Dutch or three Belgian participating centers. The study was approved by the Institutional Review Board of the Amsterdam UMC, location VUMC (ethic code: 2014.268).

### 4.2. Pharmacokinetic Sampling

VCR was administered to patients throughout treatment with a dose of either 1.5 mg/m^2^ or 2 mg/m^2^ (depending on standard care, as defined by the concerning treatment protocol and treatment phase), with a maximum of 2 mg. A push injection was administered with the required dose diluted in an injection of 10 mL 0.9% NaCl or in an intravenous bag of 50 mL 0.9% NaCl. A push injection was defined as an administration time between 1 and 5 min. The 1 h infusions were administered using an intravenous bag, in which the required dose was diluted in 50 mL of NaCl 0.9%. The infusion rate of 1 h infusions depended on the hospital. Standard administration was in 60 min (bag of 50 mL VCR in 38 min plus flushing of the VCR filled line in 22 min); however, in four patients from one hospital, total administration time was 96 min (bag of 50 mL in 60 min plus flushing of the VCR filled line in 36 min). The administration of push injections and 1 h infusions was done both through a central venous catheter, either directly (push) or extended with a line (some of the push injections (depending on the hospital) and all of the 1 h infusions). Peripheral blood was sampled either using a separately placed peripheral cannula or using the central line after it was thoroughly rinsed, according to previously published protocol [44]. Peripheral blood (2 mL) was collected in lithium heparin tubes at 10, 20, 30, 40, 60, 75, 140 and, if the child was still in the hospital, 1440 min after the start of VCR treatment. Depending on the length of the treatment protocol, samples were taken at 1–5 different occasions. 

Plasma was separated by centrifuging the sample at 4000 rpm for 5 min and subsequently stored at −80 °C until analysis.

### 4.3. Assessment of VIPN

VIPN was assessed on the same day of PK sampling using two different instruments. Of the CTCAE (version 4.03 [45]), the items’ peripheral sensory neuropathy (score 0–5), peripheral motor neuropathy (score 0–5), constipation (score 0–5), and neuralgia (score 0–3) scores were used to calculate a VIPN sum score. A CTCAE item score of two or higher was defined as VIPN. Furthermore, the Dutch translated version of the ped-mTNS [46] was used. This validated instrument, which includes both a questionnaire part (sensory, functional, and autonomic symptoms) and a physical examination, was developed to assess VIPN in children aged 5 years or older. As such, in the current study, this instrument was not used in children below 5 years of age. A total ped-mTNS score of ≥5 was defined as VIPN [46].

### 4.4. Quantification of Plasma VCR Concentrations

Vincristine sulphate (C46H56N4O10*H2SO4) was purchased from Sequoia Research Products (Pangbourne, UK). HPLC grade methanol and acetonitrile originated from Biosolve Ltd. (Amsterdam, The Netherlands). Ammonium acetate and ammonia 25% were from Merck (Darmstadt, Germany). Double distilled water was used throughout analysis.

Concentration of VCR in plasma samples were analyzed using HPLC/tandem mass spectrometry (HPLC/MS/MS), coupled with electrospray ionization (ESI), as reported previously [47]. Briefly, plasma samples of 30 µL were protein precipitated with acetonitrile/methanol (50:50, *v*/*v*), containing the internal standard vinorelbine and 7-AAD. After vortex-mixing and centrifugation, 10 µL of the supernatant was injected on the analytical column. A Finnigan TSQ Quantum Ultra triple quadrupole mass spectrometer equipped with an ESI source (Thermo Fisher Scientific, Waltham, MA, USA) operating in the positive ion mode was used as a detector. Chromatographic separation of VCR and the internal standards was carried out using a LC-20AD prominence binary solvent delivery system with a column oven, DGU-20A3 online degasser, and a SiL-HTc controller (Shimadzu, Kyoto, Japan). The mobile phase was composed of a mixture of 1 mM (70:30, *v*/*v*) ammonium acetate/acetonitrile adjusted to pH 10.5 using 25% ammonia, and mobile phase B was methanol. Gradient elution was applied at a flow rate of 0.4 mL/min through a Xbridge C18 column (502.1mm i.d., particle size 5 mm; Water, Milford, MA, USA), protected with a 0.5 mm filter (Upchurch Scientific, Oak Harbor, WA, USA), and thermostatted at 40 °C. Standards (0.25–100 ng/mL) were included in each batch of samples. HPLC run time was 6 min. The assay quantified VCR concentrations in plasma from 0.25 to 100 ng/mL. Concentrations of VCR were expressed as ng/mL. Samples were appropriately diluted and re-analyzed when the concentrations were beyond the upper limit of quantification (ULOQ) of the assay.

### 4.5. Pharmacokinetic Analysis

A linear two-compartment model with first-order elimination was used to describe the VCR concentration vs time data. Specifically, the model was fit to pharmacokinetic data from all individuals, simultaneously using non-linear mixed-effects modeling (Monolix, version 5.1.0) with the stochastic approximation expectation-maximization (SAEM) method. The PK parameters estimated included apparent Cl (L/h/m^2^)) and V_1_ (L/m^2^), along with IC-Cl (L/h/m^2^) and V_2_ (L/m^2^). The inter-individual and inter-occasion variability of the parameters was assumed to be log normally distributed. A proportional residual error model was used with assumed normal distribution of the residuals. In addition, the individual post-hoc PK estimates (empirical Bayesian estimates (EBE)) were used to estimate the VCR plasma and peripheral AUC (0–3 h), C_max_, and time above 1 ng/mL. A graphical depiction of our final model is shown in Figure 3.

### 4.6. Concurrent Azole Antifungal Treatment

Data regarding the concurrent azole antifungal treatment were collected from a medical chart review. If the child used any type of azole antifungals in the week preceding or on the day of PK measurement, this measurement was considered as performed during the concurrent azole antifungal treatment.

### 4.7. Statistical Analysis

Covariate analyses were handled either directly in the population pharmacokinetic modeling for the primary model parameters or using the two-stage approach for the secondary exposure parameters. In the population PK analysis, covariates were considered significant in a univariate analysis if their addition to the model reduced the objective function value (OFV) by at least 3.84 units (*p* < 0.05, based on the chi-square test for the difference in the −2-log-likelihood between two hierarchical models that differ by 1 degree of freedom), and the covariate term was significantly different from zero (*p*  <  0.05 by a *t*-test). The two-stage approach, used to evaluate the covariate effects on the exposure measures, was performed as follows: (1) Estimate the population PK, generate the individual post-hoc PK parameters (EBE), and calculate each occasion’s measures of exposure using its EBE; (2) use linear mixed-effects models to evaluate the effects of the covariates on the individual exposure estimates. The studied covariates using this approach were age, sex, VCR dose per m^2^, disease, concurrent azole antifungal treatment, and self-declared ancestry.

Relation between PK parameters and VIPN were analyzed using mixed effect models for continuous outcomes. Dichotomous outcomes were analyzed using logistic generalized estimating equations (GEEs). By using multivariable analysis, results were corrected for age, sex, VCR dose per m^2^, disease, concurrent azole antifungal treatment, and self-declared ancestry. Descriptive data of normally distributed variables were reported as mean ± SD and skewed variables as median and interquartile range. A two-tailed *p* value of < 0.05 was considered statistically significant. Statistical analyses regarding the relation between PK and VIPN were performed using SPSS 26.0 (Chicago, IL, USA).

## 5. Conclusions

In conclusion, the current study showed that intercompartmental clearance (IC-Cl), volume of the peripheral compartment (V_2_), and plasma and peripheral C_max_ were significantly associated with the administration method of VCR in children with cancer. By prolonging VCR infusion from push injections to 1 h infusions, VCR treatment exposure remained similar, while lowering IC-Cl. A higher IC-Cl was significantly associated with VIPN. Therefore, prolonging VCR infusions is a good strategy to maintain similar or even higher exposure of VCR, without increasing the rate of VIPN.

## Figures and Tables

**Figure 1 cancers-12-01789-f001:**
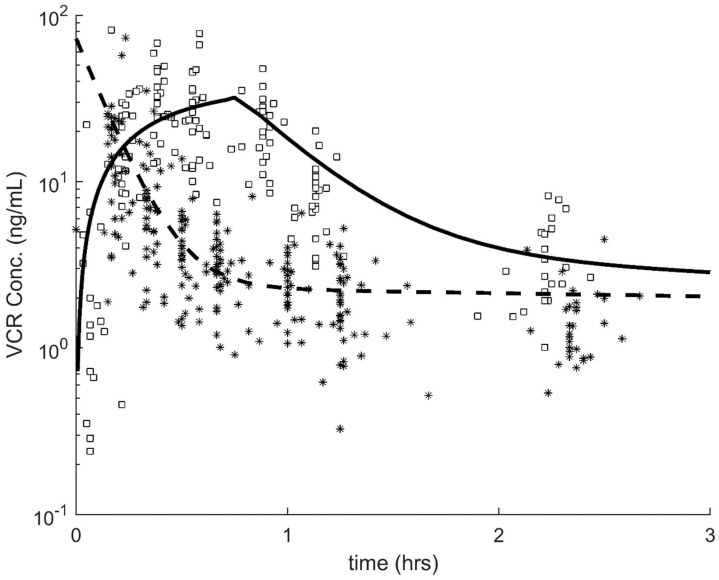
Vincristine concentration vs time plot. Black squares: 1 h infusion data; solid black curve: model fit to the 1 h infusion data; black stars: push data; dashed black curve: model fit to the push data.

**Figure 2 cancers-12-01789-f002:**
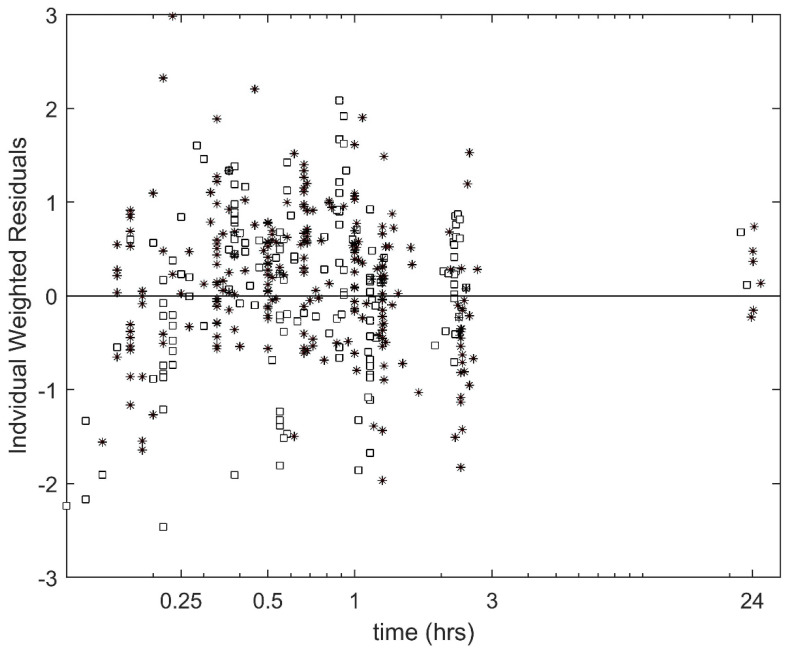
Individual weighted residuals for the final model. Black Squares: 1 h infusion data; black stars: push data.

**Figure 3 cancers-12-01789-f003:**
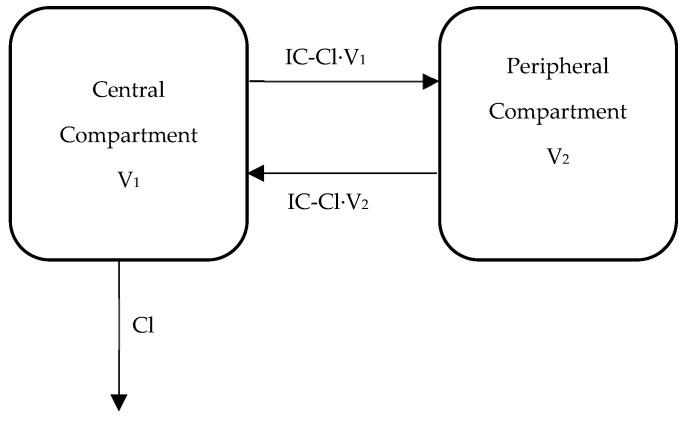
Two-compartment pharmacokinetic model. Cl: apparent clearance (L/h/m^2^)); V_1_: volume of the central compartment (L/m^2^); IC-Cl: inter-compartmental clearance (L/h/m^2^); and V_2_: volume of the peripheral compartment (L/m^2^).

**Table 1 cancers-12-01789-t001:** Patient characteristics of the included patients.

Patient Characteristics	Pharmacokinetic Part of Trial			RCT Part of Trial
	Total (*n* = 35)	Push Group (*n* = 20)	1 h Group (*n* = 15)	Total Group (*n* = 90)
**Age in years, mean (SD)**	10.06 (5.6)	10.30 (6.17)	9.73 (4.85)	9.17 (5.15)
**Ancestry, *n* (%)**				
**Caucasian**	30 (86)	17 (85)	13 (87)	73 (81)
**Non-Caucasian**	5 (14)	3 (15)	2 (13)	17 (19)
**Sex, *n* (%)**				50 (56)
**Male**	16 (45)	9 (45)	7 (47)	50 (56)
**Female**	19 (54)	11 (55)	8 (53)	40 (44)
**Diagnosis, *n* (%)**				
**ALL**	26 (74)	12 (60)	14 (93)	58 (64)
**Hodgkin**	6 (17)	5 (25)	1 (7)	18 (20)
**Medulloblastoma**	1 (3)	1 (5)	0 (0)	2 (2)
**LGG**	1 (3)	1 (5)	0 (0)	2 (2)
**Wilms**	1 (3)	1 (5)	0 (0)	8 (9)
**RMS**	0 (0)	0 (0)	0 (0)	2 (2)

Randomized clinical trial (RCT); standard deviation (SD), acute lymphoblastic leukemia (ALL), low-grade glioma (LGG).

**Table 2 cancers-12-01789-t002:** Population pharmacokinetics of vincristine.

Pharmacokinetic Parameter	Base Model		Administration Type	
	Population Estimate	RSE (%)	Population Estimate	RSE (%)
**Cl (L/hr/m^2^)**	25.3	10.1	30.5	10.0
**V_1_ (L/m^2^)**	22.1	14.4	20.8	12.5
**IC-Cl (L/hr/m^2^)**	77.9	12.8	34.2	9.5
**beta on IC-CL (push)**			1.13	4.5
**V_2_ (L/m^2^)**	358.5	11.3	127.7	19.0
**beta on V_2_ (push)**			1.13	19.1
**Inter-Individual Variability**	CV%		CV%	
**Cl (L/hr/m^2^)**	0.51	17.4	0.52	16.9
**V_1_ (L/m^2^)**	0.62	20.7	0.55	23.9
**IC-Cl (L/hr/m^2^)**	0.67	16.7	0.48	16.4
**V_2_ (L/m^2^)**	0.48	17.4	0.41	17.0
**Residual (CV%)**	0.46	4.4	0.45	4.3
**−2-Log-likelihood**	2212.5		2174.3	

Base Model: model not considering any covariates; Administration Type: model accounting for the effects of the method of administration (1 h infusion vs push) on the pharmacokinetic parameters. Relative standard error (RSE); clearance (Cl); volume of distribution of the central compartment (V_1_); inter-compartmental clearance (IC-Cl); volume of distribution of the peripheral compartment (V_2_); coefficient of variation (CV). Covariate model: PK (pharmacokinetics) = PK_pop_ * exp(beta * push), where push = 0 if 1 h infusion and 1 if push.

**Table 3 cancers-12-01789-t003:** Post-hoc pharmacokinetics of vincristine per the administration method, additionally corrected for age, sex, vincristine (VCR) dose per m^2^, disease, concurrent azole antifungal treatment, and self-declared ancestry.

Parameter	Total Group		1 h Infusions		Push Injections		Comparison of Push Injections VS. Push 1 h Infusions		
	Median	IQR	Median	IQR	Median	IQR	Beta (95% CI) *	95% CI *	*p*-Value
**Plasma AUC (ng·hr)/mL**	39.78	31.91–49.47	44.04	35.52–57.17	38.60	30.54–43.66	−2.13	−7.97 to 3.71	0.46
**Peripheral AUC (ng·hr)/mL**	36.71	27.48–45.78	42.50	33.78–55.75	35.36	26.22–40.97	−4.18	−10.26 to 1.90	0.17
**Plasma C_max_ ng/mL**	57.28	31.91–74.47	30.05	21.86–39.07	72.44	58.64–86.12	43.01	31.55 to 54.47	<0.001
**Peripheral C_max_ ng/mL**	2.87	2.24–4.37	4.81	3.15–6.94	2.57	1.77–2.92	−1.79	−2.63 to −0.94	<0.001

Interquartile range (IQR);confidence interval (CI); area under the concentration time curve (AUC); maximum concentration (C_max_). * 1 h = reference group; beta represents the difference of the corresponding parameter between the two groups.

**Table 4 cancers-12-01789-t004:** Association between pharmacokinetic parameters and vincristine induced peripheral neuropathy, additionally corrected for age, sex, VCR dose per m^2^, disease, concurrent azole-antifungal treatment, and self-declared ancestry.

	Total CTCAE Score		Total Ped-mTNS Score		VIPN CTCAE Score No/Yes *		VIPN Ped-mTNS Score No/Yes *	
	Beta (95% CI)	*p* Value	Beta (95% CI)	*p* Value	OR (95% CI)	*p*-Value	OR (95% CI)	*p*-Value
**Cl(L/hr/m^2^)**	0.00 (−0.04 to 0.05)	0.90	0.02 (−0.09 to 0.14)	0.70	1.01 (0.96–1.06)	0.74	1.05 (0.99–1.12)	0.11
**V_1_ (L/m^2^)**	−0.00 (−0.06 to 0.09)	0.96	−0.03 (−0.19 to 0.13)	0.72	1.00 (0.89–1.11)	0.93	0.98 (0.91–1.07)	0.98
**IC-Cl (L/h/m^2^)**	0.01 (0.00–0.02)	0.04	0.04 (0.01–0.07)	0.004	1.02 (1.00–1.03)	0.02	1.05 (1.02–1.09)	0.001
**V_2_ (L/m^2^)**	−0.00 (−0.00 to 0.00)	0.88	0.01 (−0.00 to 0.02)	0.20	1.00 (0.99–1.00)	0.29	1.00 (1.00–1.01)	0.81
**Plasma AUC (ng·h/mL)**	−0.01 (−0.06 to 0.03)	0.54	−0.06 (−0.16 to 0.05)	0.27	0.97 (0.93–1.02)	0.21	0.95 (0.89–1.02)	0.18
**Peripheral AUC (ng·h/mL)**	−0.01 (−0.05 to 0.04)	0.76	−0.05 (−0.16 to 0.06)	0.34	0.98 (0.93–1.03)	0.40	0.97 (0.91–1.04)	0.41
**Plasma C_max_ (ng/mL)**	−0.01 (−0.03 to 0.02)	0.55	0.00 (−0.08 to 0.07)	0.94	0.96 (0.92–1.00)	0.05	0.99 (0.96–1.02)	0.53
**Peripheral C_max_ (ng/mL)**	−0.00 (−0.36 to 0.35)	0.99	−0.31 (−1.27 to 0.65)	0.52	0.97 (0.64–1.47)	0.89	0.96 (0.61–1.49)	0.85

Common toxicity criteria of adverse event (CTCAE); pediatric-modified total neuropathy score (ped-mTNS); vincristine induced peripheral neuropathy (VIPN); clearance (Cl); confidence interval (CI); volume of distribution of the central compartment (V_1_); intercompartmental clearance (IC-Cl); volume of distribution of the peripheral compartment (V_2_); area under the concentration time curve (AUC);, maximum concentration (C_max_). * No VIPN as reference.

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
