# Peer review of "Population Pharmacokinetics of Vincristine Related to Infusion Duration and Peripheral Neuropathy in Pediatric Oncology Patients"

_cancers, 2020, doi:10.3390/cancers12071789_

Round 1

Reviewer 1 Report

General comments.

Thank you for the opportunity to review this manuscript. Dr. Velde and colleague reported the effects of administration duration (push injections versus one-hour 155 infusions) on the pharmacokinetics of VCR in children with cancer along with the effects of VCR PK 156 on VIPN. The paper provides important issue for pediatric oncological treatments. This is a well written, interesting, and useful contribution, which I think is entirely suitable for publication in Cancers.

Specific comments.

  1. Could you please tell me why you choose the toxicity, peripheral neuropathy? I suppose the other toxicity, gastrointestinal disturbance or bone marrow suppression are also important toxicity. Do you have these data?
  2. I failed to understand table 4. The author wrote some annotations (lines 148-150), but I am confusing. Please clarify that table and add the explanations.
  3. The authors should discuss about the pathogenesis of peripheral neuropathy, association between toxicity and drug exposure (total dose, concentration, time above or AUC). Which is associated with the occurrence of toxicities?

I hope that my comment is very useful for the improvement of the article.

Author Response

Reviewer #1

General comments.

Thank you for the opportunity to review this manuscript. Dr. Velde and colleague reported the effects of administration duration (push injections versus one-hour 155 infusions) on the pharmacokinetics of VCR in children with cancer along with the effects of VCR PK 156 on VIPN. The paper provides important issue for pediatric oncological treatments. This is a well written, interesting, and useful contribution, which I think is entirely suitable for publication in Cancers.

 Specific comments.

  1. Could you please tell me why you choose the toxicity, peripheral neuropathy? I suppose the other toxicity, gastrointestinal disturbance or bone marrow suppression are also important toxicity. Do you have these data?

Response: Thank you for addressing these questions. We realize that peripheral neurotoxicity is not the only type of toxicity related to vincristine (VCR). However, in clinical practice it is the main dose-limiting toxicity of VCR. That was our primary reason to make this toxicity the main objective of our study (and not to collect data on other types of toxicity). Furthermore, other chemotherapeutic agents used in the treatment of the pediatric oncology patients in general do not have peripheral neurotoxicity as a side effect, whereas other types of VCR toxicity, such as alopecia, gastro-intestinal disturbance, and bone marrow suppression, are also associated with treatment with other chemotherapeutic agents. Since all patients included in our study received a multimodal treatment regimen, it would be rather difficult to attribute for instance bone marrow suppression solely to VCR.

  1. I failed to understand table 4. The author wrote some annotations (lines 148-150), but I am confusing. Please clarify that table and add the explanations.

Response: Unfortunately, it only comes now to our attention that one of the title rows of this table was not included correctly in this table, which makes it very difficult to understand. We apologize for this. We have adapted the title of the concerning row in the revised manuscript. We have inserted an upper row to this table below line 152. In this table, we present the results of the most important pharmacokinetic (PK) outcomes in relation to the major VCR-induced peripheral neuropathy (VIPN) outcomes. These analyses were additionally adjusted for age, sex, VCR dose per m2, disease, concurrent azole-antifungal treatment, and self-declared ancestry. The beta value represents the increase in total CTCAE/ped-mTNS score per unit increase in PK outcome. For instance, if the intercompartmental clearance (IC-Cl) increases by 1, the total CTCAE score increases by 0.01. These beta values are very small, due to the small units in the different PK outcomes. The p-values represent the significance of the beta values. The odds ratio (OR) represents the difference in odds between the groups with and without VIPN according to the dichotomized outcome of either CTCAE or ped-mTNS and the corresponding p-value represents the significance of this OR. We hope that by correcting the upper row of table 4 as well as providing the additional information as described above has provided the required clarifications.

  1. The authors should discuss about the pathogenesis of peripheral neuropathy, association between toxicity and drug exposure (total dose, concentration, time above or AUC). Which is associated with the occurrence of toxicities?

Response: Thank you for asking about this. The pathogenesis of VIPN is complex. Disruption of the microtubules within the cell leads to neuronal axon dysfunction. Furthermore, inflammatory processes have been suggested as possible pathophysiologic mechanisms. In the end, this results in a progressive motor, sensory and autonomic nerve damage due to dysfunction of Aβ, Aδ and C-fibers. To elaborate on this, we have included the following lines in the fourth paragraph of the Introduction (lines 60-63): “VIPN is caused by the disruption of microtubules within the cell, resulting in neuronal axon dysfunction. Inflammatory processes also seem to play a role in the pathophysiology of VIPN. Ultimately, it leads to a progressive motor, sensory and autonomic nerve damage due to dysfunction of Aβ, Aδ and C-fibers [14].” The relation between studies PK parameters and VIPN is mentioned in paragraph 3 of the Introduction and the cited reports. Briefly, it was previously shown that the level of the main metabolite of VCR, M1, was significantly associated with VIPN according to Egbelakin et al (2011) [1]. Furthermore, Lavoie Smith et al. (2013) [2] have found the area under the concentration time curve (AUC) to be related to VIPN. All other studies investigating the effect of several PK parameters (including some of the parameters you mentioned in your question) did not show an effect on VIPN. However, as can be read in the fourth paragraph of the Discussion (lines 205-209) this could be due to lack of a proper method to assess VIPN in children prospectively.

References:

  1. Egbelakin A, Ferguson MJ, MacGill EA, Lehmann AS, Topletz AR, Quinney SK, et al. Increased risk of vincristine neurotoxicity associated with low CYP3A5 expression genotype in children with acute lymphoblastic leukemia. Pediatric Blood and Cancer. 2011;56(3):361-7.
  2. Lavoie Smith EM, Li L, Hutchinson RJ, Ho R, Burnette WB, Wells E, et al. Measuring vincristine-induced peripheral neuropathy in children with acute lymphoblastic leukemia. Cancer Nurs. 2013;36(5):E49-E60.

I hope that my comment is very useful for the improvement of the article.

Reviewer 2 Report

The quality of presentation need some amelioration, figure and table are split between two page; difficult to read.

You have to check the abbreviation, for example RCT  is written on l. 88 and describe l.230.

We need some other information about the study population, in the results part of the paper :

  • what is the dose 15 patients on 35 of the PK study did receive ?
  • number of patient with antifungal therapy

Table 4 is difficult to understand, the link between VIPN and concentration/ type of injection of VCR is incomprehensible.

Author Response

Reviewer #2

  1. The quality of presentation need some amelioration, figure and table are split between two page; difficult to read.

Response: Thank you for pointing this out. We have adapted the presentation of the Results section; all Tables and Figures are now presented on a new page and not split between two pages.

  1. You have to check the abbreviation, for example RCT is written on l. 88 and describe l.230.

Response: Thank you for this comment. We have carefully re-read the manuscript to make sure that each abbreviation is only written in full the first time it is mentioned in the manuscript.

  1. We need some other information about the study population, in the results part of the paper:
  • what is the dose 15 patients on 35 of the PK study did receive?

Response: We are not 100% sure to which 15 out of 35 patients the reviewer is referring. As mentioned in the methods section (lines 252-254): “VCR was administered to patients throughout treatment with a dose of either 1.5 mg/m2 or 2 mg/m2 (depending on standard care as defined by the concerning treatment protocol and treatment phase) with a maximum of 2 mg”. So all patients, irrespective of randomization status, received the same VCR dose. In case the reviewer is referring to “…while one-hour infusions were given in 15 patients (43%)” (line 94): these patients received the standardized dose of 1.5 mg/m2 or 2 mg/m2 with a maximum of 2 mg. Therefore, we have adapted this sentence accordingly for clarification “…while one-hour infusions were given in 15 patients (43%). All patients in the study received the standardized dose of 1.5 mg/m2 or 2 mg/m2 with a maximum of 2 mg.”

  • number of patient with antifungal therapy

Response: In lines 214-215 in the Discussion section of the manuscript it is mentioned that “… (only 7/70 measurements were with concurrent azole antifungal treatment)”. We have added to this section that these measurements were in n=6 patients, resulting in the following adaptations “… (only 7/70 measurements (n=6 patients) were with concurrent azole antifungal treatment)”.

  • Table 4 is difficult to understand, the link between VIPN and concentration/ type of injection of VCR is incomprehensible.

Response: Thank you for this remark. A comparable comment was also mentioned by the other reviewer. Unfortunately, it only comes now to our attention that one of the title rows of this table was not included correctly in this table, which makes it very difficult to understand. We apologize for this. We have adapted the title of the concerning row in the revised manuscript. We have inserted an upper row to this table below line 152. In this table, we present the results of the most important pharmacokinetic (PK) outcomes in relation to the major VCR-induced peripheral neuropathy (VIPN) outcomes. These analyses were additionally adjusted for age, sex, VCR dose per m2, disease, concurrent azole-antifungal treatment, and self-declared ancestry. The beta value represents the increase in total CTCAE/ped-mTNS score per unit increase in PK outcome. For instance, if the intercompartmental clearance (IC-Cl) increases by 1, the total CTCAE score increases by 0.01. These beta values are very small, due to the small units in the different PK outcomes. The p-values represent the significance of the beta values. The odds ratio (OR) represents the difference in odds between the groups with and without VIPN according to the dichotomized outcome of either CTCAE or ped-mTNS and the corresponding p-value represents the significance of this OR. We hope that by correcting the upper row of table 4 as well as providing the additional information as described above has provided the required clarifications.
